# Systematic Review and Meta-Analysis: Prevalence of Non-Alcoholic Fatty Liver Disease and Liver Fibrosis in Patients with Inflammatory Bowel Disease

**DOI:** 10.3390/nu15214507

**Published:** 2023-10-24

**Authors:** Pilar Navarro, Lucía Gutiérrez-Ramírez, Antonio Tejera-Muñoz, Ángel Arias, Alfredo J. Lucendo

**Affiliations:** 1Department of Gastroenterology, Hospital General de Tomelloso, Tomelloso, 13700 Ciudad Real, Spain; mpilar_ns@hotmail.com; 2Instituto de Investigación Sanitaria de Castilla-La Mancha (IDISCAM), 45004 Toledo, Spain; lgutierrezramirez@sescam.jccm.es (L.G.-R.); atejeram@sescam.jccm.es (A.T.-M.);; 3Fundación del Hospital Nacional de Parapléjicos para la Investigación y la Integración. 45007 Toledo, Spain; 4Research Unit Complejo Hospitalario La Mancha Centro, 13600 Alcázar de San Juan, Spain; 5Instituto de Investigación Sanitaria Princesa, 28006 Madrid, Spain; 6Centro de Investigación Biomédica en Red Enfermedades Hepáticas y Digestivas (CIBEREHD), 28006 Madrid, Spain

**Keywords:** metabolic liver disease, inflammatory bowel disease, Crohn´s disease, ulcerative colitis, epidemiology, risk factors, cohort studies

## Abstract

Background: Non-alcoholic fatty liver disease (NAFLD) is a common concomitant condition in patients with inflammatory bowel disease (IBD). We aim to assess the magnitude of this association. Methods: We searched MEDLINE, EMBASE and Scopus libraries for the period up to February 2023 to identify studies reporting cohorts of IBD patients in which NALFLD was evaluated. Results: Eighty-nine studies were analyzed. The overall prevalence of NAFLD was 24.4% (95%CI, 19.3–29.8) in IBD, 20.2% (18.3–22.3) in Crohn’s disease and 18.5% (16.4–20.8) for ulcerative colitis. Higher prevalence was found in male compared to female patients, in full papers compared to abstracts, and in cross-sectional studies compared to prospective and retrospective ones. The prevalence of NAFLD in IBD has increased in studies published from 2015 onwards: 23.2% (21.5–24.9) vs. 17.8% (13.2–22.9). Diagnostic methods for NAFLD determined prevalence figures, being highest in patients assessed by controlled attenuation parameter (38.8%; 33.1–44.7) compared to ultrasonography (28.5%; 23.1–34.2) or other methods. The overall prevalence of fibrosis was 16.7% (12.2–21.7) but varied greatly according to the measurement method. Conclusion: One-quarter of patients with IBD might present with NAFLD worldwide. This proportion was higher in recent studies and in those that used current diagnostic methods.

## 1. Introduction

Non-alcoholic fatty liver disease (NAFLD) is present when ≥5% of the hepatocytes become fat-filled in the absence of excessive alcohol use, infectious and other causes of toxic, autoimmune metabolic hepatitis [1]. NAFLD has become one of the most common causes of chronic liver disease [2], with incidence rates having increased more than 3-fold between 2000 and 2015 [3], and a global prevalence on the rise [4]. As a result, NAFLD now affects at least one quarter of the global population [5] and is closely linked to significant metabolic comorbidities, including obesity, insulin resistance, type 2 diabetes (T2D), hyperlipidemia, hypertension and metabolic syndrome. Beyond liver damage, NAFLD increases the risk of all-cause mortality, with cardiovascular and cancer-derived mortality being especially relevant [6,7]. 

The pathogenesis and progression of NAFLD are complex and multi-factorial and involve a “multiple hits” process where different insults converge to cause fat accumulation, inflammation and progressive liver damage [8]. Together with insulin resistance and oxidative stress, inflammation involving cytokine release and immune cell activation is a key player in advancing the disease [9]. In an attempt to reflect the heterogeneous metabolic pathogenesis and inaccuracies in terminology and definitions, the new overarching term “metabolic (dysfunction) associated fatty liver disease” or MAFLD, was recently proposed [10]. More recently, the term “metabolic dysfunction-associated steatotic liver disease” (MASLD) was proposed to define patients with liver steatosis and any other cardiometabolic risk factor [11]. 

NAFLD has been described as a common comorbidity in immune-mediated inflammatory diseases [12,13]. It includes patients with inflammatory bowel disease (IBD), where liver fibrosis and hepatocellular dysfunction are particularly prevalent [14], so much so that fatty liver disease represents the most common explanation for abnormal liver tests in patients with IBD [15,16].

Crohn’s disease (CD) and ulcerative colitis (UC), the two main forms of IBD, are immune-mediated chronic relapsing inflammatory disorders that affect at least 1 in 1000 inhabitants [17]. They also share a pathogenetic involvement of genes, environment and microbiome with MAFLD [18,19,20]. Increased gut permeability has also been associated with both NAFLD and IBD [21,22].

Despite the fact that IBD is commonly considered a debilitating disease sometimes characterized by malabsorption and weight loss, recent data indicate that the prevalence of NAFLD among IBD patients is high compared to the general population [23,24]. In addition, IBD also increases the risk of cardiovascular disease, including venous thromboembolic disease, arterial thromboembolic events, strokes and ischemic heart disease, thus enhancing the importance of controlling underlying risk factors [25,26,27]. 

Previous systematic reviews have assessed the magnitude and risk factors of the association between NAFLD and IBD. However, their results are obsolete in the face of more recent literature [24]. Furthermore, the previous literature was not exhaustively compiled [14,23], nor was the risk of bias [14,24] or clinically relevant aspects such as advanced fibrosis evaluated [24]. Finally, no review has considered the current concept of MAFLD. This particular review aims to explore systematically the prevalence, assessment tools and risk factors for NAFLD in patients with IBD, as well as to evaluate relevant clinical outcomes in this population. 

## 2. Materials and Methods

### 2.1. Study Protocol

The protocol was registered on PROSPERO (CRD42022358239); the study was reported in accordance with the Preferred Reporting for Systematic Reviews and Meta-Analysis (PRISMA) guidelines [28].

### 2.2. Selection of Studies and Search Strategy

A systematic literature search was performed independently by two authors (AJL and AA) in three major bibliographical databases (PubMed, EMBASE and Scopus) from interception until December 2022. A search update for new documents was performed in February 2023. The search was not restricted with regard to date or language of publication, study design, type of report (i.e., full paper or conference abstract) or number of patients included. 

To retrieve all published reports describing the prevalence of NAFLD in un-selected patients with IBD, we consulted the thesauri for MEDLINE (MESH) and EMBASE (EMTREE) using the following search strategy: (“fatty liver OR Non-alcoholic Fatty Liver Disease”) AND (“inflammatory bowel disease OR crohn* OR ulcerative colitis”) AND (“incidence OR prevalence OR epidemiology”) NOT (“REVIEW [Publication Type] OR SYSTEMATIC REVIEW [Publication Type])”. For the Scopus database, only free text searches with truncations were carried out (Appendix A). The search was limited to titles and abstracts. To identify additional relevant articles, a hand search of the reference lists of the related documents was performed. Three reviewers (PN, LG-R and AT-M) independently screened the database search for titles and abstracts. If any of the reviewers considered a title or an abstract might meet the study eligibility criteria, the full text of the study was retrieved. 

### 2.3. Inclusion and Exclusion Criteria

To be eligible, individual studies had to report the prevalence of NAFLD in patients with IBD confirmed by clinical, endoscopic, histological and/or radiological criteria. The diagnosis of steatosis could be based on any reliable method, including imaging [i.e., ultrasonography (US), computed tomography (CT) or magnetic resonance imaging (MRI)], liver biopsy, controlled attenuation parameter (CAP) or biochemical indexes (hepatic steatosis index — HSI or fatty liver index—FLI). Diagnostic codes for IBD or NAFLD in administrative databases were also considered. 

When available, the diagnosis of MAFLD was defined as the presence of hepatic steatosis with the presence of any one of the following three conditions: overweight/obesity, diabetes mellitus or evidence of metabolic dysregulation defined by at least two of the following: (1) waist circumference ≥ 90 cm in men and ≥80 cm in women; (2) blood pressure ≥ 130/85 mmHg or requiring specific drug treatment; (3) triglyceride levels ≥ 150 mg/dL or requiring specific drug treatment; (4) HDL cholesterol levels < 40 mg/dL for men and <50 mg/dL for women; (5) prediabetes (i.e., fasting glucose levels 100–125 mg/dL, 2-h post-load glucose levels 140 to 199 mg/dL or HbA1c 5.7–6.4%); (6) C-reactive protein (CRP) level > 2 mg/L; and (7) homeostasis model assessment (HOMA) of insulin resistance score ≥ 2.5 [29] liver fibrosis was based either on liver biopsy, on the liver stiffness measure (LSM) by transient elastography (TE) or on serum indexes, including the fibrosis-4 (FIB-4) index, the AST to platelet ratio index (APRI) APRI or NAFLD fibrosis score (NFS).

The exclusion criteria were clinical guidelines, reviews, case reports, editorials and letters. Duplicate papers or those assessing the same sample, experimental studies and studies using subsets of patient cohorts from previously published research by the same group of authors were also excluded. If needed, the study authors were contacted for clarification.

### 2.4. Data Extraction

Three reviewers (PN, LG-R and AT-M) independently extracted relevant information from each eligible study using a standardized data extraction sheet and then cross-checked the results. Discrepancies were resolved by consensus. The data extracted included the last name of the first author, publication year, study location (country), study design, sample size of IBD cohort, number of subjects by sex (if available), IBD subtype (CD or UC, if available), diagnostic criteria used for the IBD and disease activity (active/remission), when available. Details on the diagnostic method for NAFLD, number of NAFLD patients in the IBD cohort, the number of patients with advanced liver fibrosis in IBD patients with NAFLD, and number of patients fulfilling MAFLD criteria were included whenever available. Disagreements between reviewers regarding data extraction were resolved through discussion or consulting a senior author (AA and AJL).

### 2.5. Risk of Bias Assessment

The risk of bias in each document was checked against the Joanna Briggs Institute Critical Appraisal Checklist for Studies Reporting Prevalence Data [30]. A study was considered to be at low risk for bias if each of the bias items could be categorized as low risk. Studies were judged to have a high risk of bias; however, if any one of the items was deemed high risk. The five investigators (AJL, AA, PN, AT-M and LG) independently gave each eligible study an overall rating of high, low or unclear risk of bias; disagreements were resolved by consensus.

### 2.6. Study Outcomes and Statistical Analysis

Summary estimates, along with their 95% confidence intervals (CIs), were calculated for the prevalence of NAFLD among IBD patients with fixed- or random-effects meta-analyses weighted for the inverse variance following DerSimonian and Laird’s Method [31]. Prevalence in male and female patients (where reported) was also estimated using proportions and with 95% CIs.

For the primary outcomes, planned subgroup analyses were performed based on type of IBD, patient sex, document type (full paper or abstract), study design (prospective, retrospective, cross-sectional), diagnostic method for NAFLD or fibrosis, geographical origin of paper and temporal trends. 

Inconsistency between studies was assessed by means of a Chi square test (Cochran Q statistic) and quantified with the I2 statistic. Generally, I2 was used to evaluate the level of heterogeneity, assigning the categories low, moderate and high to I2 values of 25%, 50%, and 75%, respectively [32]. Publication bias was evaluated with the aid of a funnel plot, the asymmetry of which was assessed through Begg-Mazumda’s rank test [33] or Egger test [34]. All calculations were performed with StatsDirect statistical software version 2.7.9 (StatsDirect Ltd., Cheshire, UK).

## 3. Results

### 3.1. Literature Search Results and Characteristics of Included Documents

Our search strategy overall produced a total of 818 documents; after removing duplicates, a total of 739 documents remained. After examining the title and abstract, 599 documents were excluded due to non-fulfillment of inclusion criteria. This yielded 140 potentially relevant documents, with 7 additional ones identified after reference tracking (Appendix A). Finally, 89 documents — 39 full papers [35,36,37,38,39,40,41,42,43,44,45,46,47,48,49,50,51,52,53,54,55,56,57,58,59,60,61,62,63,64,65,66,67,68,69,70,71,72,73] and 50 abstracts [74,75,76,77,78,79,80,81,82,83,84,85,86,87,88,89,90,91,92,93,94,95,96,97,98,99,100,101,102,103,104,105,106,107,108,109,110,111,112,113,114,115,116,117,118,119,120,121,122] — involving 1,387,184 people with IBD from 27 different countries worldwide met our study inclusion criteria. The sample size of IBD cohorts varied between 20 and 552,887 patients. All documents were published between 2010 and 2023, with the exception of two papers published in 1971 [58,73].

A total of 32 studies were carried out in European countries (Italy [37,41,52,53,57,59,67,83,84,96,116], Spain [40,61,105,109], Portugal [36,75,118], United Kingdom [58,70,73], Romania [77,86,93], Germany [45,69], Bulgaria [76], Croatia [85], Greece [98], Poland [43], Slovakia [49] and the Netherlands) [68]; seven in East Mediterranean countries (Turkey [48,78,92,110,123] Tunisia [79] and Egypt [55]), one in West Asia (Qatar) [74] and eight in South Asia (India) [95,97,100,106,107,111,119,121]. Overall, 32 studies were carried out in the Americas (including USA [35,39,42,44,51,54,56,60,65,66,80,82,87,88,90,91,94,99,101,102,103,112,114,115,117,122], Canada [38,63,113], Brazil [64,104], and Mexico) [71]. Six studies came from the Western Pacific Region (China [46,50], Japan [62], South Korea [47], Taiwan [72], and Australia [120]) and three more from North Africa (Morocco) [81,89,108]. 

Details on individual studies, including the size of IBD cohorts and diagnostic methods for NAFLD and liver fibrosis, are shown in Appendix A; the prevalence of NAFLD and fibrosis in IBD cohorts is shown in Appendix A.

### 3.2. Risk of Bias and Quality Assessment

The application of the Joanna Briggs Institute Critical Appraisal Checklist for Studies Reporting Prevalence Data revealed that 44 of the 89 studies presented a high risk of bias in at least one of the domains evaluated, with Domain 4 (detailed description of study subjects and setting; n = 31 studies) [37,74,75,76,78,79,81,82,83,84,86,89,95,98,99,100,101,102,103,104,105,106,107,108,109,110,111,112,119,121,122], being the most common, followed by Domain 3 (adequacy of the sample size; n = 16 studies) [39,42,51,55,67,75,77,79,87,89,97,103,104,106,109,119], Domain 1 (appropriateness of the sampling frame to address the target population; n = 11 studies) [37,38,39,40,50,53,55,61,102,104,108], Domain 6 (use of valid methods for the identification of the condition; n = 4 studies) [74,75,119,122] and Domain 7 (measurement of the condition in a standardized and reliable way for all participants; n = 4 studies) [44,58,60,73]. Appendix A provides details of the risk of bias for all studies included in our systematic review. 

### 3.3. Characteristics of IBD Patient Cohorts 

Only 24 out of the 89 documents—19 full-papers [37,38,40,41,42,43,45,46,47,52,53,55,59,61,62,63,65,67,68] and 5 abstracts [76,77,101,105,113]—reported on disease activity in IBD patients at assessment of NAFLD. These included 11 studies involving patients with CD [37,40,41,45,46,49,52,53,62,65,76] and 9 with UC [37,40,41,45,52,53,55,76,77]. Disease activity was mostly measured by clinical indexes; in the case of CD, they consisted of the Harvey-Bradsaw index [124] used in 9 studies [38,40,45,53,59,61,63,68,113], the Crohn’s Disease Activity Index (CDAI) [125] in 8 studies [37,40,41,46,47,52,62,76], and the pediatric Crohn’s disease activity index (PCDAI) [126] used in 2 studies [42,43].

In UC patients, the partial Mayo score [127] was used in 12 studies [38,40,41,47,52,53,59,61,63,68,76,113], and one study [45] used the simple clinical colitis activity index (SCCAI) [128]. Finally, the pediatric ulcerative colitis activity index (PUCAI) [129] was used in two studies [42,43]. 

Endoscopic scores, including the SES-CD [130] for CD, the Mayo full score [131] for UC and the Rutgeers’ score [132] for post-surgical endoscopic recurrences at ileocolic anastomosis, were used to assess IBD activity in only two studies [55,67].

### 3.4. Diagnosis Modalities for NAFLD and Fibrosis in IBD Patients

To diagnose NAFLD, 28 studies used ultrasound [37,40,41,43,45,49,50,52,55,59,61,62,71,76,78,79,82,84,86,89,92,96,97,104,106,108,109,123], five studies used CT scans [47,57,65,87,94] and four studies using MRI imaging [42,46,54,103]; four further studies based NAFLD diagnosis on non-specified imaging techniques [44,66,98,102].

Biochemical indexes, including the FLI (n = 2) [36,109], HSI (n = 7) [36,38,83,85,109,116], or NASH FibroSure^®^ (n = 1) [51], were used to define NAFLD in 10 studies overall. CAP was used for the diagnosis of NAFLD in 17 studies [36,48,53,63,67,68,69,72,77,83,95,105,107,109,113,118,121]. 

In four studies, IBD patients underwent histopathological assessment to diagnose NAFLD after altered liver function testing [39,58,73,75]. Multiple methods were used to evaluate NAFLD in five studies [60,101,110,117,133].

10 studies did not define their diagnostic method but took NAFLD diagnoses from administrative health records [35,90,91,114] or based them on IDC diagnostic codes. [56,70,80,88,99,112]. Finally, there was no method described in eight studies [64,74,81,93,100,119,120,122].

Liver fibrosis was assessed in 28 individual studies. In most of these (18), LSM by TE [40,52,53,61,63,67,68,69,72,96,97,105,109,113,116,118,120,121] was performed. Only seven studies provided an average LSM in the patient cohort, however, but not the proportion of patients with significant or advanced fibrosis. They were not meta-analyzed therefore [40,53,67,96,97,109,116]. Biochemical indexes including FIB-4 [38,54,65,85,102,123], NFS [60,117] or NASH FibroSure^®^ [51] were used in six, two and one studies, respectively. Patients underwent liver biopsy after pathological TE findings in one study [61]. One additional study evaluated historical liver biopsy records in patients with altered liver imaging or function tests [71].

### 3.5. Pooled Prevalence of NAFLD in Patients with IBD Overall and According to IBD Subtype and Sex 

Based on the results of 68 studies, the global pooled prevalence of NAFLD in patients with IBD overall was 24.4% (95% CI, 19.3–29.8; I2 = 99.7%). Differences were noted according to study design, with prevalence being higher in cross-sectional studies (32.5%; CI, 7.9–64.1; I2 = 99.7%) compared to retrospective (23.2%, CI, 17.2–29.9; I2 = 99.8%) and prospective (24.3%; CI, 13.6–377; I2 = 99%). Studies reported as full-text papers also provided a higher prevalence (27.5%; CI, 20.3–35.3; I2 = 99.3) than those exclusively reported as conference abstracts (22.1%; CI, 15.2–29.9; I2 = 99.7). Details are provided in Table 1 and Appendix A. The prevalence of NAFLD in patients with CD was provided in 46 studies, which were pooled to provide an overall prevalence of 20.2% (95%CI, 18.3–22.3; I2 = 99.7). In patients with UC, the pooled prevalence of NAFLD in 41 studies was 18.5% (95%CI, 16.4–20.8; I2 = 99.5%) (Appendix A). As in IBD overall, cross-sectional studies and full-text papers provided higher pooled prevalence rates in both CD and UC-restricted studies. 

Significant publication bias was found for studies reporting on the prevalence of NAFLD in IBD overall, CD and UC patients in funnel plot analysis. Analysis of funnel plot symmetry arising from heterogeneity suggests the existence of distinct subgroups of studies, each with a different intervention effect (Appendix A) [134].

There were 19 studies that reported the prevalence of NAFLD according to patients’ sex and specified the numbers of patients in each cohort. These included 15 studies in patients with IBD in general [36,38,42,45,47,52,59,60,61,63,69,72,80,92,99], 7 studies in patients with CD [35,45,46,54,62,79,121], and one study in patients with UC [45]. The prevalence of NAFLD among males with IBD was 28.0% (CI, 14.5–44.0; I2 = 99.5), whereas in females it was 22.5% (CI, 11.6–35.9; I2 = 99.4). In patients with CD, the pooled prevalence of NAFLD was also higher in males (27.5%; CI, 9.1–51.2; I2 = 99.2) compared to female patients (24.2%; CI, 7.0–47.6; I2 = 98.8). 

### 3.6. Pooled Prevalence of NAFLD in Patients with IBD According to Diagnostic Method of NAFLD

We next analyzed the prevalence of NAFLD according to the modality used for diagnosis in patients with IBD. The highest prevalence was found in IBD patients assessed by CAP (38.8%; 95%CI, 33.1–44.7; I^2^= 90.3; n = 17 studies), followed by liver biopsy (30.7%; 95%CI, 22.8–39.1; I^2^ = 47.1; n = 4 studies) and HIS (30.4%; 95%CI, 26.1–35; I^2^ = 66.8; n = 6 studies).

Among diagnostic methods based on imaging techniques, US provided the highest prevalence of NAFLD in IBD patients (28.5%; 95%CI, 23.1–34.2; I^2^ = 97.4%; n = 28 studies), with MRI (n = 4 studies) and CT-scan (n = 5 studies) providing NAFLD prevalence figures of 25.1% (95%CI, 13.1–39.5; I^2^ = 93.8) and 23.2% (95%CI, 7.6–44.3; I^2^ = 98.6), respectively. Appendix A and Figure 1 provide details on the prevalence of NAFLD in patients suffering from IBD according to the diagnostic method used. No significant publication bias was found according to funnel plot symmetry and the Begg test (Appendix A).

### 3.7. Prevalence of Fibrosis in Patients with IBD and NAFLD According to the Fibrosis Assessment Tool

The prevalence of liver fibrosis (significant or advanced, as defined by each source study and assessed by any method) in patients with IBD who presented with NAFLD was provided in 20 studies and was 16.7% (95%CI, 12.2–21.7; I^2^ = 88.9). In the 11 studies that used TE to define liver fibrosis, the summary estimate for fibrosis prevalence was 23.6% (95%CI, 17.4–30.4; I^2^ = 83.7). The prevalence of liver fibrosis measured with biochemical indexes was lower at 14.2% (95%CI, 8.2–21.4; I^2^ = 83.9%) from the results of the six studies that used FIB-4, and only 3.7% (95%CI, 2–5.3) in the two studies that used NFS (Appendix A and Figure 2). Again, no significant publication bias was found according to funnel plot symmetry and the Begg test for studies assessing liver fibrosis by any method or by TE (Appendix A).

### 3.8. Variations in NAFLD Prevalence in Patients with IBD According to World Region and Temporal Trends

Geographical differences were noted in the prevalence of NAFLD among IBD patients according to the world region. The pooled prevalence of NAFLD among IBD patients was 31.8% (95%CI, 22.1–42.4; I^2^ = 99.4) in European countries (n = 32 studies); 27.7% (95%CI, 19.5–36.8; I^2^ = 96.9) in East Mediterranean countries (n = 10 studies) and 14.2% (95%CI, 12.5–16; I^2^ = 99.8; in the Americas (n = 32 studies). 

The summary estimate of NAFLD prevalence in IBD patients in South Asia was 19.7% (95%CI, 10.8–30.4; I^2^ = 93.8; n = 8 studies) and in the western Pacific region 18.7% (95%CI, 12.2–26.2; I^2^ = 90.7; n = 5 studies). 

In order to provide current prevalence estimates, we shortened the results of studies using CAP to define NAFLD in patients with IBD. After pooling results from 10 studies, the prevalence in Europe was 43.1% (95%CI, 34.3–52.1; I^2^ = 92.3); 35.7% (95%CI, 30.1–41.5; n = 2 studies) in the Americas, and 28.6% (16.95%CI, 1–43; I^2^ = 87.4; n = 3 studies) in South Asia (Figure 3).

As for temporal trends, a relevant increase in the prevalence of NAFLD in patients with IBD was noted when studies published up to 2014 (17.8%, 95%CI, 13.2–22.9; I^2^ = 98.2; n = 19 studies) were compared to those published in and after 2015, the year NAFLD was included as a MeSH term in PubMed (23.2%; 95%CI, 21.5–24.9; I^2^ = 99.7; n = 70 studies).

### 3.9. Prevalence of MAFLD in Patients with IBD

Only one study [61] specifically included patients with IBD diagnosed with MALFD according to recently proposed criteria [8]: on the basis of the presence of hepatic steatosis in patients with BMI ≥ 25 kg/m^2^, T2D, or evidence of metabolic dysregulation [29]. In this study, the prevalence of MAFLD was 42% in the IBD population overall, with no differences in CD or UC patients (42.7% and 41–4%, respectively). 

We attempted to estimate how many patients with IBD could have fulfilled criteria for MALFD among the available IBD cohorts reporting for NAFLD: Overall, 18 studies informed on the prevalence of obesity/being overweight among patients with IBD and NAFDL [35,36,38,42,44,47,52,54,60,62,66,72,80,90,91,99,112,115], which ranged between 2.8% and 75%. Twenty-one studies reported the prevalence of T2D in IBD patients with NAFLD [35,36,38,39,44,45,47,52,54,59,61,63,66,72,80,87,90,91,115,117,121], which ranged between 1.4% and 59%. The prevalence of high blood pressure in IBD patients presenting NAFLD was reported in 16 studies [35,36,38,39,44,54,59,61,63,66,80,90,91,99,115,117] and hypercholesterolemia in 8 studies [46,47,52,54,66,86,90,91]. Increased waist circumference [59], hypertriglyceridemia [35] and increased serum C-reactive protein levels [80] in IBD patients presenting with NAFLD were each reported in one study. No study specifically reported on the prevalence of pre-diabetes or insulin resistance (Appendix A). With these data, we estimated that the pooled prevalence of MAFLD in patients with IBD and NAFDL was at least 30.4% (95%CI, 21.9–39.6; I^2^ = 96.2%; n = 19 studies). The proportion of patients with CD and NAFLD who also presented with MAFLD could be at least 40% (95%CI, 25.4–55.5; I^2^ = 99.4; n = 5) and for UC patients, a minimum of 53.6% (95%CI, 13.8–90.8; I^2^ = 99.5; n = 4 studies) (Appendix A). No significant publication bias was found in the funnel plot symmetry analysis of MAFLD prevalence in IBD and its subtypes (Appendix A). 

## 4. Discussion

In this study, we undertook an exhaustive literature search and systematically reviewed all the available evidence on the prevalence of NAFLD in patients with IBD. After pooling the results of 39 full papers and 50 conference abstracts, spanning 27 countries worldwide and mostly published across 2 decades, NAFDL (widely variable) was found to affect around 1/4 of IBD patients overall. This is irrespective of disease type, which varied significantly when subgroup analyses were carried out. 

There are a small number of previously published meta-analyses on this subject [14,23,24], but to the best of our knowledge, the present systematic review represents the most ambitious and comprehensive attempt to assess the prevalence of NAFLD in patients with IBD. This is not only due to the retrieval of many documents not previously included in other systematic reviews (twice the number of studies compared to the most recent review) [23], but also due to the fact that eligibility criteria were not restricted to language of publication [14] or pre-defined sample size [23]. Furthermore, in order to provide an accurate representation of the standard population with IBD, patient cohorts that were not pre-selected by disease phenotype, activity, or therapy used were included. In addition, the risk of bias in each source document was appraised in detail, and prevalence figures from every source study were assessed, sorted by the criteria used by their authors to define NAFLD [24]. The evaluation of liver fibrosis, the main clinical consequence of NAFLD, was also rigorously addressed.

Our results show a summary NAFLD prevalence of 24.4% overall, similar, albeit slightly inferior, to that reported in previous systematic reviews (which ranged from 27.5% [24] to 32%) [14]. When the studies were published as full-text papers and those with a cross-sectional design were considered, prevalence summaries were slightly higher. When recently incorporated and more precise diagnostic tools were used to identify NAFLD, in comparison with other alternatives, the summary prevalence of NAFLD in patients with IBD was 38.8% in the 17 studies that used CAP. We have documented the predicted prevalence increase, which is higher in the countries with greater socioeconomic development. These results suggest that the prevalence of NAFLD in patients with IBD is increasing and could reach figures similar to those reported for the general population [5]. However, some relevant questions still remain, especially given the wide heterogeneity of the diagnostic methods used for NAFLD and its main clinical consequence, advanced fibrosis, as well as the, until now, unreported high risk of bias detected in many studies. This study shows the wide heterogeneity with which NAFLD is diagnosed (whether defined by this term or other clinical equivalents) in patients with IBD, and its contribution to the imprecision of results, and the wide inconsistency of summary estimates provided. Indeed, some studies did not mention the diagnostic criteria used to define this disease. Subgroup analyses did allow for this heterogeneity to be addressed, and to show that the prevalence of NAFLD was higher in full-text published studies, in those with more rigorous designs, and in the investigations conducted more recently. However, some inconsistencies persist, especially if we consider that the prevalence of NAFLD was lower in studies that used screening methods based on biochemical scores (HSI or FLI) or ultrasound than those based on new and more sensitive techniques, such as CAP [135]. Cut-off points used in the different studies to define hepatic steatosis using CAP also varied, and included 216 [67], 236 [48,77,113,121], 248 [63,68,69,72,105,118], 260 [36,95], 275 [109] and 288 dB/m [83,116]. Future studies should unify diagnostic criteria, and, following EASL recommendations [136], accordingly adopt the optimal CAP threshold of 275 dB/m. 

The prevalence of fibrosis similarly varied widely among studies, and when those using TE were summarized, it was present in 23.6% of patients with IBD. These data should be viewed cautiously however, as liver stiffness cut-off values used to define fibrosis were broadly variable among studies, and consisted in ≥6 [97], ≥7 [52,63,67,69,121], ≥7.3 [61,68], or ≥8 kPa [40,72,105,113,118,120], thus involving the same patients who would have been classified differently according to the study in which they were included. MAFLD is a more recent concept proposed to compensate for the conceptual inaccuracies of the conventional NAFLD tag. MAFLD mainly focuses on the presence of metabolic dysfunction among subjects with a fatty liver, excluding those with metabolically uncomplicated fatty liver [29]. Compared to NAFLD, MAFLD was able to predict better cardiovascular risk [137], and atherosclerotic cardiovascular disease [138] in asymptomatic subjects. However, until now the prevalence of MAFLD among IBD patients has not been addressed in a systematic review. Based on our review of the literature, at least 30.4% of patients with IBD who also suffer from NAFLD would fulfil criteria for MAFLD. This figure most likely underestimates the real magnitude of the problem, as it was found to affect 42% of IBD patients, irrespective of whether they had CD or UC, in the single study found that addressed this association [61], and which contradicts another study’s findings of metabolic affectation being restricted to patients with CD [139]. More recently, a multi-society consensus proposed MASLD as a novel nomenclature and diagnostic criteria in order to improve awareness and patient identification [11]. However, no studies have used this term in epidemiological studies involving IBD patients.

Our work has the limitation that it is based on studies that are not bias free; only 45 were considered to be of low or unclear risk of bias, with the rest presenting issues due to insufficient or inadequate study subjects and setting description, and/or a limited or inappropriate sampling frame to reliably assess the IBD population. In addition, some studies may well have used unreliable or nonstandard methods to identify or determine NAFLD in some patients. The fact that the majority of studies included in our review were abstracts, restricted full assessment of their methodology and results, and thus contributed to this evaluation.

Unlike previous systematic reviews [14,23], we have not attempted to identify factors associated with risk of NAFLD in patients with IBD, (other than the already expected higher prevalence in males), but to find out the frequency of NAFLD in patients with IBD from observational studies. The search strategy we used was not designed to identify risk factors, so some relevant studies may have been missed. Similarly, we did not assess the relationship between IBD characteristics (disease type, phenotype, extension, activity or therapy) and risk for NAFLD, since very few studies provided accurate information on these aspects. To find how IBD interacts with cardiometabolic factors to promote NAFLD and its consequences should therefore be addressed in a further investigation. 

## 5. Conclusions

The results of this meta-analysis, undertaken on 89 studies mostly published in the last two decades and involving over 1.3 million people, indicate that approximately one-quarter of patients with IBD could experience NAFLD worldwide, with trends increasing and with recent novel technology-based research providing a prevalence of 38.8%. Despite abundant literature on the subject, the high risk of bias in most of the studies means that our results need to be taken cautiously. There is clearly a need for new, better-designed studies to help determine the contribution of different demographic, clinical, and therapeutic variables, and their interaction with well-known metabolic risk factors, to the development of NALFD and liver dysfunction in patients with IBD. In this way we will be able to identify patients at risk and design personalized effective interventions.

## Figures and Tables

**Figure 1 nutrients-15-04507-f001:**
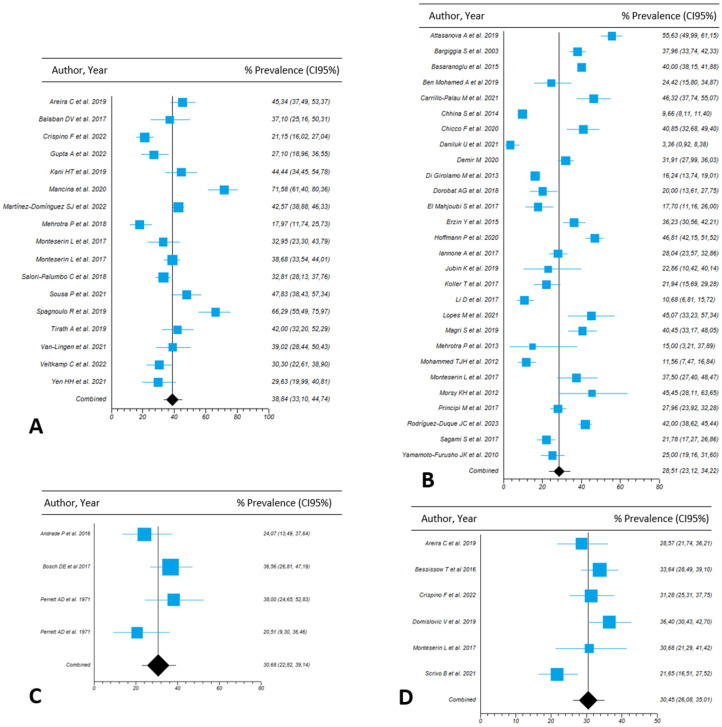
Forest plots of the pooled prevalence of non-alcoholic fatty liver disease in patients with inflammatory bowel disease according to diagnostic methods used, including (**A**) controlled attenuation parameter (CAP) measure; (**B**) liver ultrasound; (**C**) liver biopsy; and (**D**) hepatitis steatosis index (HSI) score.

**Figure 2 nutrients-15-04507-f002:**
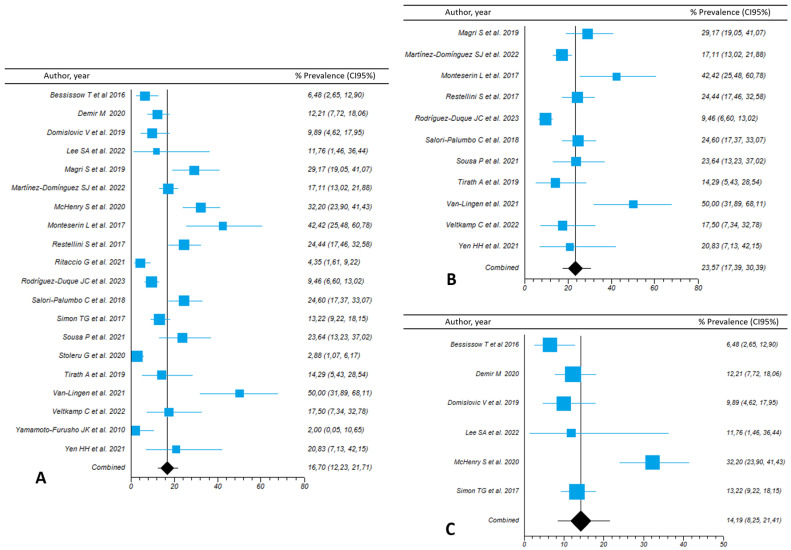
Forest plots of the pooled prevalence of liver fibrosis in patients with inflammatory bowel disease overall (**A**), independently of the method of assessment used. Studies using transient elastography (**B**) and biochemical indexes (**C**) are shown separately.

**Figure 3 nutrients-15-04507-f003:**
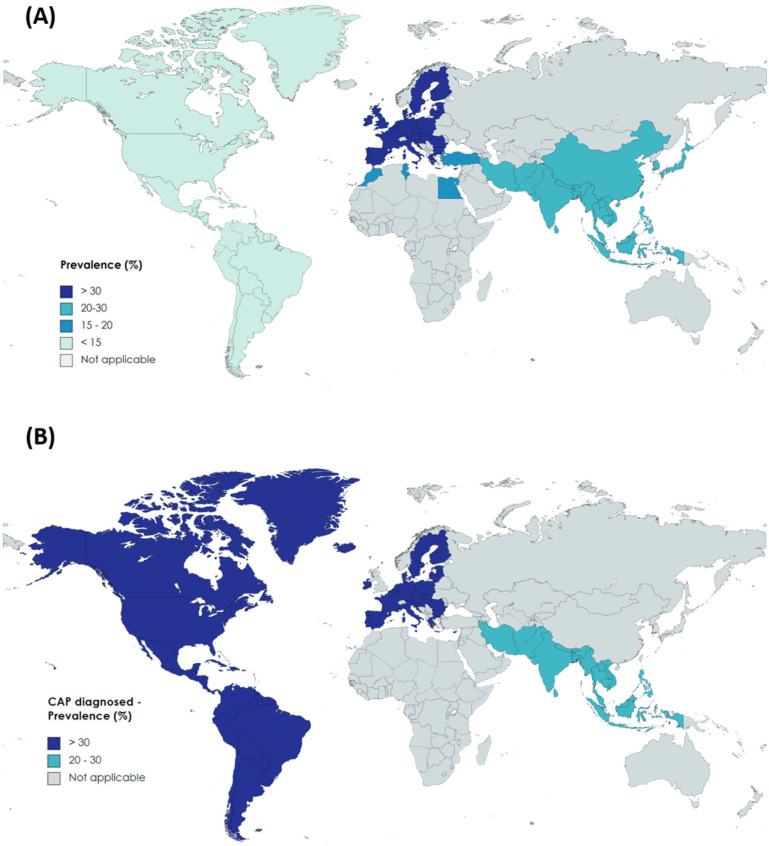
Prevalence of non-alcoholic fatty liver disease in patients with inflammatory bowel disease worldwide, according to World Health Organization regions. In (**A**) the result of polled prevalence of 89 studies mostly published along 2 decades; in (**B**) studies that used controlled attenuation parameter or CAP, as a diagnostic method, published in the 2017–2022 period.

**Table 1 nutrients-15-04507-t001:** Pooled prevalence of non-alcoholic fatty liver disease (NAFLD) in patients with inflammatory bowel disease (IBD) overall, Crohn’s disease and ulcerative colitis according to sex, type of publication and study design. Prevalence of metabolic (dysfunction-) associated fatty liver disease (MAFLD) in the same groups of patients.

		Inflammatory Bowel Disease			Crohn’s Disease			Ulcerative Colitis		
		n	Proportion (95%CI)	I^2^	n	Proportion (95%CI)	I^2^	n	Proportion (95%CI)	I^2^
NAFLD		68	24.4 (19.3–29.8)	99.7	46	20.2 (18.30–22.27)	99.7	41	18.52 (16.36–20.79)	99.5
	Male	15	28 (14.5–44)	99.5	7	27.5 (9.1–51.2)	99.2	-	-	-
	Female	15	22.5 (11.6–35.9)	99.4	7	24.2 (7–47.6)	98.8	-	-	-
	Full paper	29	27.5 (20.3–35.3)	99.3	26	26.2 (19.0–34.1)	98.7	23	24 (16.1–32.9)	99
	Abstract	39	22.1 (15.2–29.9)	99.7	20	12.7 (10.9–14.6)	99.7	18	12.4 (10.4–14.4)	99.5
	Prospective	15	24.3 (13.6–37.7)	99	7	22 (6.4–4.4)	98.8	10	19.9 (9.2–33.5)	97.7
	Retrospective	46	23.2 (17.2–29.9)	99.8	32	19 (16.9–21.3)	99.7	25	17.3 (14.9–19.9)	99.6
	Cross-sectional	7	32.5 (7.9–64.1)	99.7	7	25.6 (4.6–56.1)	99.5	6	23.6 (3.4–54.2)	99.5
MAFLD		19	30.4 (21.9–39.6)	96.2	9	40 (25.4–55.5)	99.4	4	53.6 (13.8–90.8)	99.5

## Data Availability

Not applicable.

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
