# Peer review of "Systematic Review and Meta-Analysis: Prevalence of Non-Alcoholic Fatty Liver Disease and Liver Fibrosis in Patients with Inflammatory Bowel Disease"

_nutrients, 2023, doi:10.3390/nu15214507_

Round 1

Reviewer 1 Report

Non-alcoholic fatty liver disease (NAFLD) is a common concomitant condition in patients with inflammatory bowel disease (IBD), the authors included eighty-nine studies and assessed the magnitude of this association. The overall writing of the article is ok, but there are the following points that need to be revised.

1. Although a search update for new documents was performed in February 2023, it need to update once again.

2.This systematic review included full papers and 50 abstracts, why so many abstracts included, why didn't you find the full text and how to extract the data?

3. The results of each meta are very heterogeneous, which will seriously affect the authenticity of the results. Please explain the reasons for the heterogeneity in detail and do subgroup analysis and sensitivity analysis.

4,The discussion is not deep enough, so it is suggested to further discuss the different differences, which are different from those of this study.

Author Response

Non-alcoholic fatty liver disease (NAFLD) is a common concomitant condition in patients with inflammatory bowel disease (IBD), the authors included eighty-nine studies and assessed the magnitude of this association. The overall writing of the article is ok, but there are the following points that need to be revised.

We appreciate the effort made by this reviewer in assessing our manuscript

  1. Although a search update for new documents was performed in February 2023, it need to update once again.

Thank you for the suggestion. Since the date our last update of the literature searches, in a PubMed search we found three potential documents to be considered in this systematic review, all reporting original data. The first one (Onwuzo S et al. Cureus 2023;15:e35854) did not assess the prevalence of NAFLD directly, but that of potential risk factors for NAFLD, so it should not be considered. The second one (Kyung Hyun H et al. Gut Liver 2023 May 19. doi: 10.5009/gnl220409) defined NAFLD and fibrosis in terms of biochemical indexes (which are no diagnostic methods, but screening tools, so results provided are likely not accurate enough. The third paper (by McHenry et al. Inflamm Bowel Dis 2023: izad129. doi: 10.1093/ibd/izad129) provides accurate data which could be certainly included in our systematic review. However, they derived from a patient series of 363 individual patients, who will not modify the result of our study, obtained from a series of 1.3 million patients. For information, the Journal of Crohn’s & Colitis has just published a meta-analysis on hepatobiliary manifestations in patients with IBD, carried out on 118 studies (a number very similar to ours), retrieved from a search completed in May 2020.

2.This systematic review included full papers and 50 abstracts, why so many abstracts included, why didn't you find the full text and how to extract the data?

We performed an extensive literature research, in major databases for documents leading with the PICO question we tried to answer with this study. Many of the documents retrieved were conference abstract (this is, summaries presented to congresses) which never followed by a proper full-text article. We believe having retrieved and evaluated such as amount of “grey literature” is a strength of our research, and never a limitation. For the remaining documents, the full text was available and could be read full-text to extract data. 

  1. The results of each meta are very heterogeneous, which will seriously affect the authenticity of the results. Please explain the reasons for the heterogeneity in detail and do subgroup analysis and sensitivity analysis.

The reviewer noted the wide heterogeneity of data included in the meta-analysis we performed, which resulted in high inconsistency, according to I2 statistic. We humble believe this mathematical finding that do not put into question that our results are authentic, but they derived from somehow heterogeneous studies that are affected by wide variability among the results of individual source studies. In order to address such high inconsistence, we performed different subgroup analysis, which aimed to avoid combinations of certain studies which used different criteria to define endpoints (this is, fatty liver defined by results of ultrasonography, biochemical indexes or controlled attenuation parameter, etc.). This way we minimized the risk that the summary effects may ignore important differences between studies.

  1. The discussion is not deep enough, so it is suggested to further discuss the different differences, which are different from those of this study.

Thank you for the suggestion. We preferred to devote the major extent of the manuscript to presenting the extensive results of our meta-analysis, which also required an abundant supplementary material. The structure and content of the Discussion was defined according to the essential points stablished by the PRISMA statement.  

Reviewer 2 Report

The design, concept and idea of the paper are solid. There is one metodological issue that could be improved. It is not the best approach to mix the objective methods like imaging with biochemical indexes in eligibility of the studies. Results were presented in a manner where methods were compared correctly, so this initial problem was not aggravated. Nevertheless, it might be more appropriate not to include studies that diagnosed liver steatosis exclusively using FLI.

Other issue that I find quite important is the fact that the new terminology was not even mentioned. Since the concept is not new probably there are no studies on IBD and MASLD. Nevertheless, the new approach and terminology should be descibed in the introduction and discussion -A multi-society Delphi consensus statement on new fatty liver disease nomenclature - ScienceDirect

Author Response

The design, concept and idea of the paper are solid. There is one metodological issue that could be improved. It is not the best approach to mix the objective methods like imaging with biochemical indexes in eligibility of the studies. Results were presented in a manner where methods were compared correctly, so this initial problem was not aggravated. Nevertheless, it might be more appropriate not to include studies that diagnosed liver steatosis exclusively using FLI.

We appreciate the comments made by this reviewer. In fact, we performed several subgroup analyses that are described in detail across the manuscript’s results section, in order to reduce clinical variability in the description of study endpoints when they were assayed by variable (and not necessarily overlapping methods).  By doing such, we put into question the accuracy of the data already provided by many studies in this topic.

Other issue that I find quite important is the fact that the new terminology was not even mentioned. Since the concept is not new probably there are no studies on IBD and MASLD. Nevertheless, the new approach and terminology should be descibed in the introduction and discussion -A multi-society Delphi consensus statement on new fatty liver disease nomenclature - ScienceDirect

Thank you for your comment. We have mention the novel nomenclature suggested for the reviewer and included a new reference to the paper indicated. However, this change nothing about the scientific content of the manuscript, as no document retrieved for the systematic review that supports our study has considered this new nomenclature system.